# Operational Decisions and Sustainability: A Brazilian Case of a Drugs Distribution Center

**João Thiago de G. A. A. Campos** *,†,‡ 🄓, **Renato de Castro Vivas** †,‡ 🄓,
**Adonias Magdiel Silva Ferreira** †,‡ **and Francisco Gaudêncio Mendonça Freires** †,‡

Escola Politécnica - Dpto. de Engenharia Mecânica, Federal University of Bahia, Salvador 40170-110, Brazil;
renato.vivas@ufba.br (R.d.C.V.); adoniasmagdiel@ufba.br (A.M.S.F.); francisco.gaudencio@ufba.br (F.G.M.F.)
* Correspondence: jcampos@ufba.br; Tel.: +55-7199-2954-687
† Current address: Aristidis Novis, 2—Federação, Salvador 40170-110, Brazil.
‡ These authors contributed equally to this work.

**Abstract:** Recently, the supply chain in the pharmaceutical sector, which is important economically to the healthcare industry worldwide, has received special attention owing to different factors involved in the distribution of drugs. Furthermore, it has an important role in global sustainability as organizations base their efficient decisions on the results from performance analysis of economic indicators. Thus, the sustainability of operations decisions must be analyzed to achieve better decision efficiency. This study integrates analytical methods of operational activities evaluation for a drug distribution center in a pharmaceutical logistic organization to analyze the sustainability of its operations. Furthermore, a proposed framework incorporates time variability management (TVM) decisions into a trade-off analysis of triple bottom-line (TBL) sustainability dimensions and operations managers' decisions. The framework is a real-time data-gathering decision system that evaluates processes using stochastic simulation and process efficacity based on control-chart analysis and analyzes the trade-off performance. Managers' decisions on time variability is modeled using an Analytical Hierarchy Process. The results of the trade-off analysis of sustainability and TVM indicate that economic dimensions have a higher impact on an organization than social and environmental dimensions. Managers assume that social and environmental impacts are less important to organizations' performance. Environmental and social dimensions have different impacts on time variability decisions, where managers assume that operations' time reduction has more impact on the social dimension, while operations' time increase has a higher environmental impact. Thus, the framework is an effective tool for analyzing the sustainability of operations decisions, which is associated with variability analysis.

**Keywords:** sustainability; time variability; trade-off; decision-support system

## 1. Introduction

Supply chain management is one of the most important cost-driven decisions in organizations [1]. Based on economic analysis, supply chain disruptions also have social and environmental impacts [2]. Thus, a sustainability approach is necessary to convert better management decisions into a decision-making process. Specifically, in pharmaceutical supply chains, operational management of supplying drugs to customers is highly sensitive to organization performance [3]. Furthermore, evaluation of the sustainability of operations in the pharmaceutical supply chain is necessary to support decision-making models and increase organization efficiency.

In terms of sustainability, some authors analyze the organizational performance based on the triple-bottom-line (TBL) dimensions of organizational sustainability, such as economic, social,

and environmental [4–6]. Ref. [5] proposes the integration of TBL into a visual business canvas, to explore allowing sustainability in business innovations. In terms of a strategic view, ref. [4] analyzes a macrodimension using four indices to assess sustainability performance. However, ref. [6] reveals that strategic sustainability models are insufficient for providing a satisfactory framework for sustainable politics owing to the lack of connection between several aspects that affect the TBL dimensions. Thus, the basis of a sustainable organization is measured by performance dimensions, which consider that sustainable organizations may achieve high performance and better results, such as profitability, social care, and environment preservation. Generally, sustainability is analyzed to support strategic decisions, but operational management also requires tools to improve their decisions in day-to-day operations and their impact on sustainability factors [7]. Strategic levels and sustainability studies are widely explored. Risk management in strategic level and sustainability is proposed by [8], where the authors conceptualize a framework to examine the effects of sustainability, reporting practices, and business performance. Ref. [9] associate financial performance and sustainability with managerial and operational capabilities, suggesting sustainability strategies to be their promoter. Ref. [10] integrates sustainability into strategic decision-making by using a fuzzy AHP method to select the sustainability issues most relevant to strategic decisions. However, operational studies and their relationship with the economy, society, and environment require more attention. Considering the operational management sensibility of supply chains and the importance of sustainability in organizations, the integration of these areas is necessary to help in the decision-making process [11,12]. Furthermore, in sustainability, which includes a wide range of organizational performance indicators, the decision-making trade-off should be analyzed for the improvement of organizations [13]. Integrating the trade-off relationship into a decision-support system (DSS) can also improve decision-making [14–16], specifically when sustainability performance is evaluated.

Thus, to incorporate an operational perspective in sustainability and trade-off analysis, which is still unexplored in academic research, this study proposes a trade-off analysis of sustainability TBL dimensions such as economic, social, and environmental performance that affect operational management decisions. The case of a Brazilian drug distribution center is analyzed and evaluated in this study using a decision model that uses an analytical hierarchy process (AHP), and a trade-off framework evaluation is made. Furthermore, the time variability management (TVM) of operations is used as a condition of the decision model.

Under a general approach, methods for analyzing sustainability use quantitative and qualitative data. In quantitative analysis, ref. [17] reviews a large number of relevant papers that use quantitative methods to identify gaps and future research perspectives. Refs. [18,19] are more specific in their sustainability analysis that is integrated with the supply chain using a proposed framework, and they collaborate to understand the evolution of studies in sustainable supply chain management (SSCM). Furthermore, refs. [20,21] propose multi-objective models. Qualitative methods are also well defined in academic literature [1,22–24]. To achieve more effective results, some authors explore the integration of multi-criteria and multi-objective decision methods [25–29]. In strategic, tactical, and operational performance dimensions, applying methods to ensure effectiveness of performance dimensions is a common practice. For example, ref. [30] analyzes the operations performance by costs key performance indicator. Even costs are generally associated with strategic management, the author associates it directly to operations management. Ref. [31] analyzes the Indian pharmaceutical industry through the information system performance in supply chain management operations. Ref. [32] also regards the operations management by an information system approach, which explores the cross-functional teams alignment and the operational effectiveness. Ref. [33] analyzes information management by examining the role of information in the internal as well as external context of process management and operations performance. However, the integration of sustainability performance measurements into operations management remains unexplored. Although operations management has been widely studied [34–36], the studies are not related to sustainability. Some researchers have conducted studies on operations management and sustainability. Ref. [13] have developed a research synthesis about state-of-the-art

empirical studies relating to the impact of sustainability practices on organizations with respect to economic, social, and environmental dimensions. Results from this study demonstrate positive effects of sustainability on costs, product quality, and mix flexbility based on operational measurements. Ref. [37] worked on knowledge management to enable sustainability in organizational operations and to integrate sustainable operations into business strategies. The evaluation of operations performance and its relationship with information and knowledge comprise the main theme of this study. For example, management philosophies such as lean manufacturing and total quality management have several tools to improve operational activities. For example, ref. [38] assess the impact of total quality management on environmental performance. In terms of a management approach, sustainability and operations have focused on supply chain management. Ref. [39] associate social sustainability with supply chain practices and its impact on operational performance. The authors emphasize that the effect of sustainability orientation on operational performance is significantly moderated. However, the integration of sustainability performance measurements to organization operations remains unexplored.

Specifically, for TVM, the operation time is generally evaluated by quantitative and qualitative methods based on statistical models using stochastic and deterministic analyses. TVM is a widely used method to reduce variability of operations time in main management philosophies such as six sigma and lean manufacturing. However, traditional management behaviors do not consider trade-off relations in operations-based decisions [40,41]. In particular, operational activities and sustainability analysis are effective techniques for the analysis of operations variability with a trade-off decision model between TBL dimensions of sustainability. To achieve process efficiency at the operational level, a stochastic model is defined and, further, simulated through scenario comparisons [42–47]. A more realistic approach to decision-support systems in operational management levels is determined when TVM is used to reduce process variability and to evaluate the impact of sustainability in an organization. Ref. [48] analyzes and defines modeling techniques and their integration based on data type, thereby providing a robust model that allows operational managers to make more effective decisions. In particular, the SSCM of drug distribution should be analyzed to provide a decision-making model for managers that can improve the efficiency of the pharmaceutical industry. Thus, a framework is proposed and validated by using the operational management decisions and the TBL sustainability performance of a Brazilian drug distribution center.

The study is divided into the following sections. The proposed framework section presents the methods to collect, organize and analyze data with the help of an exploratory data analysis. Thereafter, a process analysis has been conducted with the assistance of a control chart as well as a TVM and sustainability trade-off analysis that shows the trade-off method. Next, the framework validation and case study section presents the results of applying the proposed framework and discusses. The conclusion section presents the discussion of sustainability and operational performance analysis of the case study as well as the managerial implications.

## 2. Proposed Framework

A framework to integrate TVM and sustainability impacts in a drug distribution center is proposed in this study. Based on traditional management decisions, the TBL trade-off is evaluated based on time variability reduction decisions related to operations. TVM forecasting in real time can validate the proposal and may be expanded to other operations-based decisions, such as preventing the wastage of resources. Considering the variability responses from resources use, the framework can also guide managers to more effective and efficient decisions, owing to forecast analysis and the impact of variability reduction decisions on organization performance. Variability is a consequence of unbalanced systems, where each operation has a critical impact on all performance dimensions of the organization [49,50].

Then, by using TVM and sustainability integration framework, a trade-off analysis is proposed based on four premises, which consolidate the decision-making framework:

1.  Bottom-up decisions are made from the operations-based level and are used for higher levels according to their needs.
2.  The variability behavior of parameters is the operation time of each operator, and the TBL performance is analyzed using this DSS tool.
3.  Output of this framework results in a trade-off analysis that supports the decision-making process of the operational management level.
4.  The framework allows the use of any mathematical model for trade-off analysis; in this study, the AHP is used.

The DSS tool analyzes the trade-off behavior of management decisions in specific steps, as shown in Figure 1:

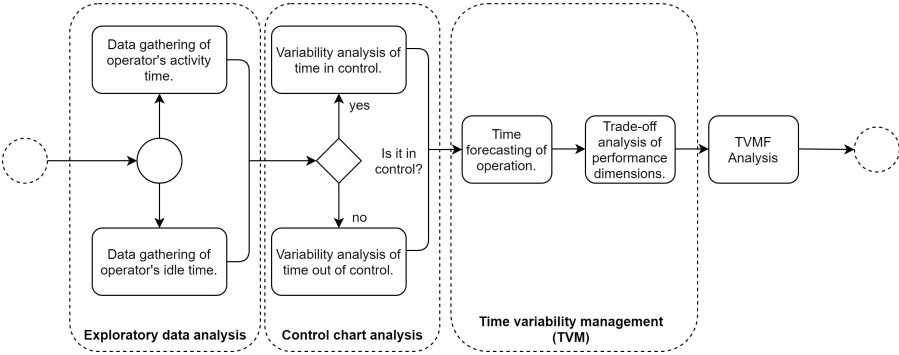

**Figure 1.** Proposed framework of Time Variability Management.

The data analysis is divided into three main steps: exploratory data analysis, control-chart analysis, and TVM. It also provides trade-off indicators based on forecasted operator behavior and the impact over the TBL dimension performance defined by the AHP weights.

*2.1. Exploratory Data Analysis*

Operations of the DSS use a continuous time collecting method, where

$$Q_{eio} = \sum_{i=0}^{1} T_{eio} - T_{(e-1)io}.$$
(1)

Calculated by a continuous elapsed time evaluation *T*, the procedure for collecting the sample time that is used as an input for the framework starts when the operator obtains the receipt drug box and ends when he or she inserts the box back on the conveyor. A process is a systematic group of activities. Then, the process time *P* is calculated using Equation (2).

$$P = \sum_{1}^{o} \sum_{e=1}^{n} \sum_{i=0}^{1} (T_{eio} - T_{(e-1)io}),$$
(2)

where:
*T*: continuous elapsed time,
*e*: measured time index,
*i*: measured time type (activity or idleness) index,
*o*: total operations for a single operator, and
*n*: sample size.

Further, a probability distribution characterization is performed through a chi-squared statistical distribution analysis of the sample. This process is followed by a quartile analysis (to identify outliers) and interval estimation of mean parameter to calculate the sample representability:

$$\chi^2 = \sum_{all\ classes} \frac{(Q_{eio} - v_{eio})^2}{v_{eio}}, \tag{3}$$

where:
$v$: expected time into chi-squared distribution.

The number of degrees of freedom is the sample size, which in this research is 100 events per operation. A hypothesis test is performed to verify whether the sample fits to the distribution probability curve:

$H_0$: observed values are equal to the theoretical values and
$H_a$: observed values are not equal to the theoretical values .

A likelihood ratio test, which is used to evaluate the hypotheses, is defined as

$$LRT = -2\ln(\frac{L(\hat{\theta}_0\chi)}{L(\hat{\theta}_A\chi)}), \tag{4}$$

where $\hat{\theta}_0$ and $\hat{\theta}_A$ are the maximum likelihood estimation of each distribution. The null hypothesis is accepted if the p-value test is higher than 0.05.

### 2.2. Control Chart and Time Variability Analysis

A control chart is generated based on the three-sigma control limits:

$$\begin{aligned} UCL &= \mu + 3(\sigma_{\overline{Q}eio}), \\ LCL &= \mu - 3(\sigma_{\overline{Q}io}). \end{aligned} \tag{5}$$

Using the central limit theorem, where $\overline{Q}_{io}$ is normally distributed, the upper control limit (UCL) and the lower control limit (LCL) are defined as the boundaries of the trusted interval. In this study, the three-sigma control limits defined by [51] are considered.

Then, a variability analysis is performed by mapping, modeling, and forecasting the time variability of each operator.

***Mapping and modeling.*** A Petri net model is used to simulate the behavior of the operator. The graphical and mathematical basis of this well-consolidated method is well defined [52–54].

Based on literature, ref. [52] defines the Petri net as a 5-tuple, i.e., $PN = (P, T, F, W, M_0)$,

where:
$P = \{p_1, p_2, \ldots, p_m\}$ is a finite set of places,
$T = \{t_1, t_2, \ldots, t_n\}$ is a finite set of transitions,
$F \subseteq (P \times T) \cup (T \times P)$ is a set of arcs (flow relation), and
$W : F \rightarrow \{1, 2, 3, \ldots\}$ is a weight function,
$P \cap T = \varnothing$, and $P \cup T \neq \varnothing$.

Thus, the process map is a 3-tuple, $M = (P, T, F)$, represented by a generic $C$ matrix of places and transitions:

$$
C = \begin{matrix} & \begin{matrix} T_1 & T_2 & \dots & T_n \end{matrix} \\ \begin{matrix} P_1 \\ P_2 \\ \vdots \\ P_m \end{matrix} & \begin{pmatrix} W_{11} & W_{12} & \dots & W_{1n} \\ W_{21}0 & W_{22} & \dots & W_{2n} \\ \vdots & \vdots & \vdots & \vdots \\ W_{mn} & W_{mn} & \dots & W_{mn} \end{pmatrix} \end{matrix}. \tag{6}
$$

To simulate the discrete event, the Petri net must comply with the firing rule [52], as expressed in (7).

$$
\forall p \subset P, M' = M + C(., t). \tag{7}
$$

Furthermore, the DSS generates a TVM factor by calculating the coefficient of variation and by multiplying it by the operator time variability in Equation (8):

$$
TVMF = \begin{matrix} \text{Idle Time} \\ \text{Activity Time} \end{matrix} \begin{matrix} \text{In control} & \text{Out of control} \\ \begin{pmatrix} CV_{11} * f(Q_{e0o}) & CV12 * f(Q_{e0o}) \\ CV_{21} * f(Q_{e0o}) & CV_{22} * f(Q_{e0o}) \end{pmatrix} \end{matrix}, \tag{8}
$$

where:
$CV$ : coefficient of variation and
$TVMF$ : time variability management factor (TVMF).

The maximum value of time variability between activity or idle time is then selected for TVMF in Equation (8), which is modeled into the equation system (9).

$$
f(x) = \begin{cases} \sup(100 * \frac{s_{e0o}}{x_{e0o}} * \frac{\mu_{e0o} - \overline{Q_{e0o}}}{\mu_{e0o}}; 100 * \frac{s_{e1o}}{x_{e1o}} * \frac{\mu_{e1o} - \overline{Q_{e1o}}}{\mu_{e1o} - LCL}) \text{ if } x \leq UCL \text{ or } x \geq LCL, \\ \sup(100 * \frac{s_{e0o}}{x_{e0o}} * \frac{Q_{e0o} - \mu_{e0o}}{UCL - \mu_{e0o}}; 100 * \frac{s_{e1o}}{x_{e1o}} * \frac{Q_{e1o} - \mu_{e1o}}{UCL - \mu_{e1o}}) \text{ if } x > UCL, \\ \sup(100 * \frac{s_{e0o}}{x_{e0o}} * \frac{\overline{Q_{e0o}}}{\mu - Q_{e0o}}; 100 * \frac{s_{e1o}}{x_{e1o}} * \frac{\overline{Q_{e1o}}}{\mu - Q_{e1o}}) \text{ if } x < LCL. \end{cases} \tag{9}
$$

## 2.3. TVM and Sustainability Trade-Off Analysis

To integrate TVM and sustainability performance, an AHP method is used. The scores presented in Table 1 are the results of the AHP analysis. Although the framework allows the insertion of any mathematical model into a real-time interaction of trade-off analysis scores, for this study, the AHP analysis, which generates a constant weight, is used to validate the proposed DSS.

Then, a trade-off analysis is performed based on the proposed framework, considering the following criteria defined in AHP:

The AHP is a paired comparison ratio, derived from a relative scale of judgment or data [55]. For sustainability comparison, the criteria defined are presented in Figure 2.

The objective of an AHP analysis is to evaluate the impact of operation time variability over TBL sustainability dimensions such as economic, social, and environmental performance. For economic evaluation, it considers sustainability impact over production rate, human error operation, and equipment failure. For social evaluation, it considers operator fatigue and accident risk in operations. For environmental evaluation, energy consumption and gas emissions have been considered. Thus, TBL has been incorporated into the framework with reference to the application of the AHP method as described below.

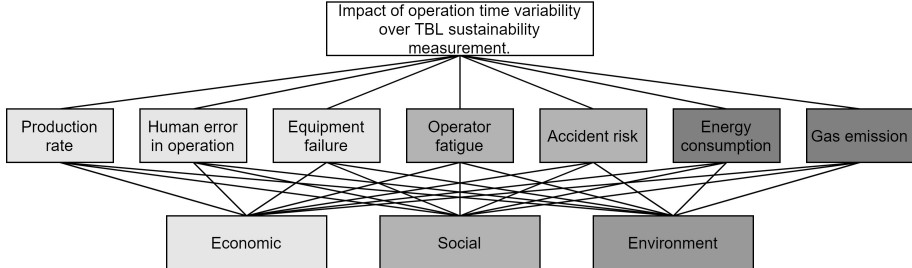

**Figure 2.** Analytical Hierarchy Process structure for sustainability performance.

Following the proposed analysis of eigenvalue and eigenvector used in a matrix calculation, ref. [29] describe a simple model, which is adapted for this research:

Assuming that there are $d_m(m = 1, ..., M)$ decision makers, on operations activities in the drug distribution center, who evaluate the performance of $n(j = 1, ..., J)$ of TBL based on $c(q = 1, ..., Q)$ criteria such as described in Figure 2, a pairwise decision matrix is formulated in Equation (10).

$$\sum_{m=1}^{M} \sum_{q=1}^{Q} c = \begin{pmatrix} & j_1 & j_2 & \dots & J \\ j_1 & 1 & p_1/p2 & \dots & p_1/p_J \\ j_2 & p_2/p_1 & 1 & \dots & p_2/p_J \\ \vdots & \vdots & \vdots & 1 & \vdots \\ J & p_J/p_1 & p_J/p_2 & \dots & 1 \end{pmatrix} \quad (10)$$

where:
$p$: the performance of each criteria by a pairwise comparison.

Furthermore, the AHP assumption analyzes a consistency ratio (CR) to evaluate the judgments of specialists (i.e., decision makers) on the criteria. Refs. [29,55] provide details of the AHP model validation and the application of their mathematical models.

Subsequently, constant trade-off scores are generated, as presented in Table 1, through a criterion weight analysis of operator behaviors. The framework output is the trade-off score presented in Table 1, based on the following rules:

- If increasing the amount of idle time is necessary to reduce variability, then the trade-off score of the upward arrow column is considered in the trade-off analysis.
- If decreasing the amount of idle time is necessary to reduce variability, then the trade-off score of the downward arrow column is considered in the trade-off analysis.

**Table 1.** Trade-off scores.

|  | Idle Time ↑ | Idle Time ↓ | Activity Time ↑ | Activity Time ↓ |
|---|---|---|---|---|
| Economic | 0.7 | 0.6 | 0.61 | 0.43 |
| Social | 0.2 | 0.2 | 0.1 | 0.247 |
| Environment | 0.1 | 0.2 | 0.143 | 0.486 |

To validate the responses of specialists, a CR is evaluated, as presented in Table 2.

**Table 2.** Operator time variability reduction consistency ratio.

| Operator Behavior | Consistency Ratio |
|---|---|
| Operator time reduction | 4.8 |
| Operator time increase | 2.2 |

A radar graph of trade-off decisions on the economic, environmental, and social performance is generated as the performance indicator of the decision-making system of the organization.

## 3. Framework Validation and Case Study

### 3.1. Process Characterization

The sustainability analysis was applied in a customer orders drugs separation section (CODSS) of a Brazilian pharmaceutical drug distribution center in Brazil. The CODSS is divided in two tasks, drug list verification and drug separation. A real-time data gathering was executed for a sample of 100 events for each task, by measuring the idle and activity times elapsed from a unique operator in process. The process is presented in Figure 3 and then modeled using the Petri net tool, which is characterized by places and transition tables.

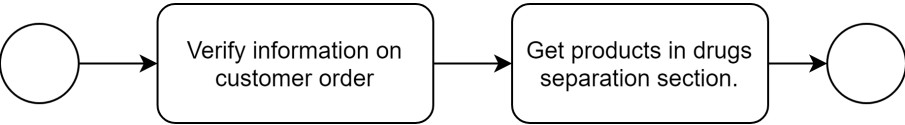

**Figure 3.** Drugs separation process map.

Each task of CODSS is modeled by a pair of place and transition. The Petri net is presented in Figure 4 and described in places and transitions description tables (displayed in Tables 3 and 4, respectively).

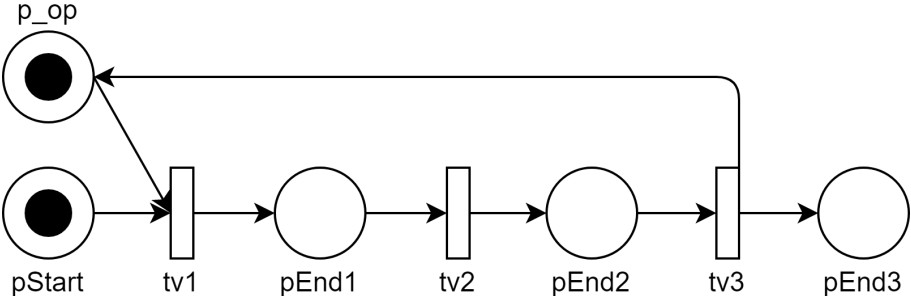

**Figure 4.** Drug separation process map.

**Table 3.** Description table for places.

| Name | Places | Description |
| --- | --- | --- |
| Process start | pStart | Customer order list arrived in drug separation section |
| Customer list | pEnd1 | Customer order is available to operator. |
| Drug picking process requested | pEnd2 | Validated drug list is available to the operator. |
| Drugs separated | pEnd3 | Drugs requested by customer are separated and ready for the next process. |
| Operator status | p_op | Operators are available for the next request. |

**Table 4.** Description table for transitions.

| Name | Transitions | Description |
| --- | --- | --- |
| Transition 1 | *tv1* | Send customer request order to drug separation section |
| Transition 2 | *tv2* | Operator verifies customer request |
| Transition 3 | *tv3* | Operator separates drugs from list |

The modeled process is represented by matrix C in Equation (11), where the weights defined represent the dynamic interactions for discrete-event process simulation, following the Petri net firing rules [52].

$$
C= \begin{array}{c} \\ pStart \\ pEnd1 \\ pEnd2 \\ pEnd3 \\ p\_op \end{array}
\begin{array}{ccc} tv1 & tv2 & tv3 \\ \left( \begin{array}{ccc} 1 & 0 & 0 \\ -1 & 1 & 0 \\ 0 & -1 & 1 \\ 0 & 0 & 0 \\ 1 & 0 & -1 \end{array} \right) \end{array}, \tag{11}
$$

A statistical model of each task is defined by a probability distribution selection of drug list verification (*tv2*) and drug separation (*tv3*) activities. Thus, the stochastic behavior is defined by a probability distribution fit, represented by the distribution name and their respective parameters. In this study, in a sample of 100 events for each transition, *tv2* and *tv3* are evaluated. *Tv1* represents the arrival of customer requests and is not considered for the evaluation of the sustainability impact. Tables 5 and 6 show the probability distribution names, probability parameters, and log likelihood hypothesis test.

**Table 5.** Activity-time probability distribution parameters.

| Process | Probability Function | Parameters | Log Likelihood |
|:---:|:---:|:---:|:---:|
| *tv2* | Gama | 1.415; 3.829 | $-1.063 \times 10^2$ |
| *tv3* | Normal | $1.588 \times 10^1$; 7.603 | $-1.379 \times 10^2$ |

**Table 6.** Idle time probability distribution parameters.

| Process | Probability Function | Parameters | Log Likelihood |
|:---:|:---:|:---:|:---:|
| *tv2* | Gamma | 5.086; 1.037 | $-88.008$ |
| *tv3* | Gammal | 78.18; $7.134 \times 10^{-2}$ | $-38.15$ |

For *tv2* and *tv3*, gamma and normal distributions were validated by the *p*-value of the log likelihood test.

### 3.2. Process Simulation and Scenario Analysis

The process simulation is necessary to identify unbalanced system problems as bottlenecks or disruptions. In this case, the process is simulated with one operator to perform *tv2* and *tv3* tasks. This process is a sequential task and is presented in Figure 3. Then, the real and simulated times are compared in a chart control, for both idle time and activity time. First, a Petri net simulation is executed using the GPENSIM MatLab library, and the graphical representation of the Petri net model states is shown in Figure 5.

In the figure, each place is described dynamically by the number of tokens. A token represents the change of state in the Petri net model, and then when each time the operator executes a task, a new state is created, and a token is moved following the Petri net rules. As described in Table 3, *pStart* denotes the start of the process, whereas *pEnd1* provides the customer drug request for the operator. In this study, the limit is 200 tokens (customer drug requests). Then, *pEnd2* represents the place of the first task modeled (*tv2*), whereas *pEnd3* represents the place of the second task of the process (*tv3*). Furthermore, to simulate the model with one unique operator, a place called *pOp* is created. This place is necessary to grant the sequential execution of tasks by a unique operator, similar to the model displayed in Figure 4.

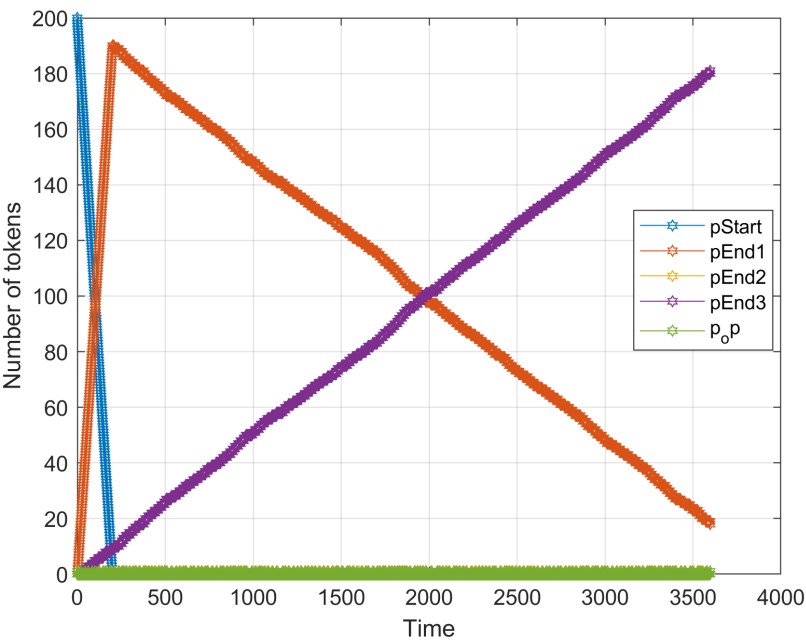

**Figure 5.** Petri net simulation using GPENSIM module.

Hence, the proposed framework simulates operator behavior. To evaluate the quality of the process, a control chart is defined to allow real-time and simulated-time analysis.

For task 1, Figures 6 and 7 represent the control chart of idle time and activity time, respectively.

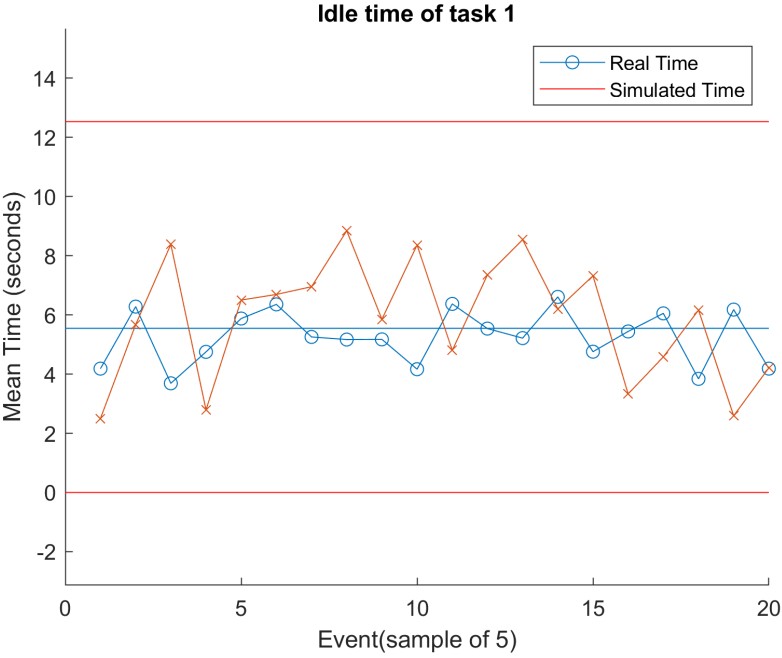

**Figure 6.** Control chart of idle time in task 1 (*tv2*).

Tasks 1 and 2 are the controlled processes. The deviation from the mean of the simulated time of both tasks is higher owing to the probability distribution fitting. However, as the simulation is based on random numbers following a probability distribution, the error is acceptable.

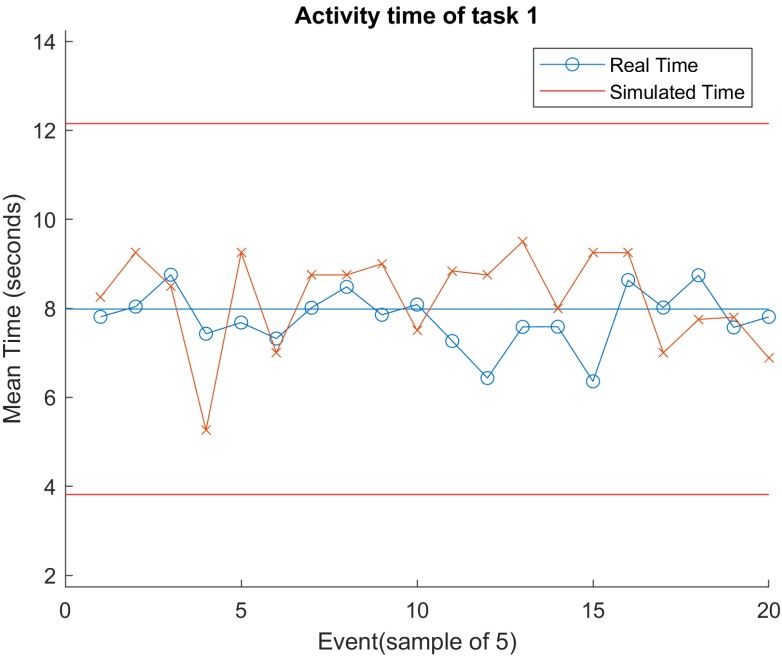

**Figure 7.** Control chart of activity time in task 1 (*tv2*).

For task 2, the real time and simulated time have lower deviations from mean, indicating that it reduces the forecasting operator behavior error and, thus, increases the effectiveness of the trade-off analysis. For the activity and idle time of task 2 represented in Figures 8 and 9, although the process is controlled, the simulated time has higher deviation and must be analyzed by managers to identify possible problems of an unbalanced system. The proposed framework is based on simulated time, and the forecasting step of this method identifies possible process management improvement points.

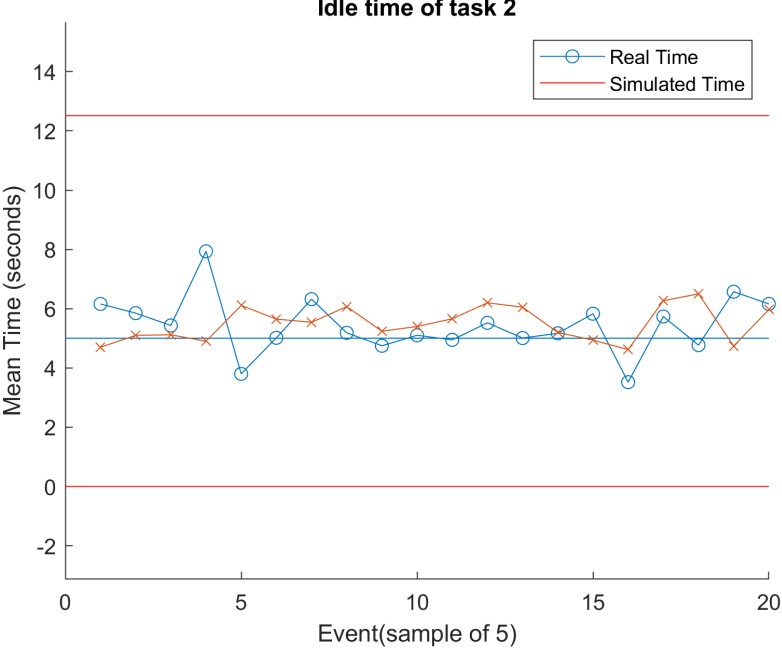

**Figure 8.** Control chart of idle time in task 2 (*tv3*).

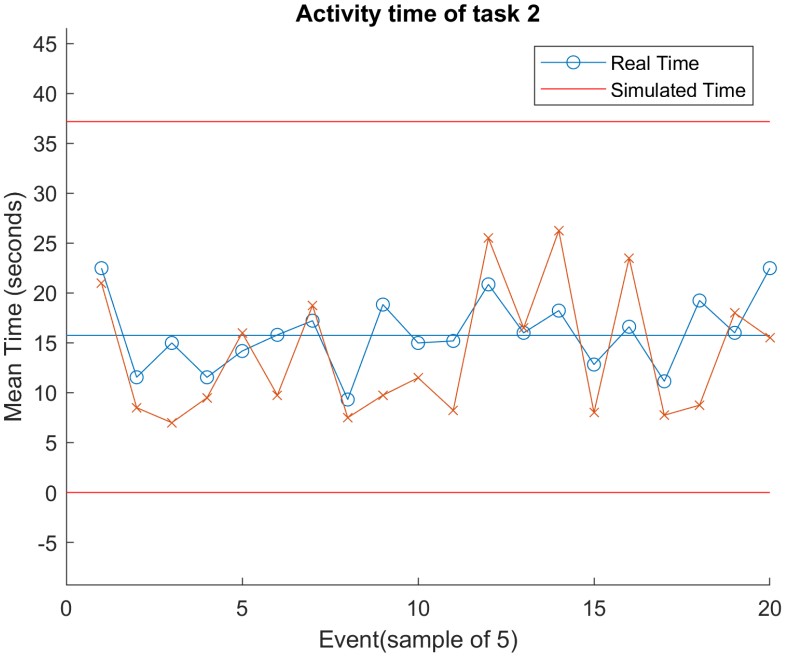

**Figure 9.** Control chart of Activity time in task 2 (*tv3*).

The control-chart analysis is an important step for the proposal of this framework because it allows managers to assess when a process is in or out of control. In CODSS, the process is well-controlled. A sustainability trade-off analysis of the manager decisions model is presented in Figures 10 and 11.

First, this research proposes to analyze the decision over TVM and the impact on TBL sustainability dimensions. In both graphs, the trade-off analysis clearly shows that economic dimensions have a higher sensitivity to TVM decisions. Generally, the productivity, quality, and maintenance dimensions are priorities for managers, which is reflected in this analysis by a higher score in the economic dimension. However, the social and environmental dimensions have different behaviors for each task. For task 1, social dimensions are more affected than environmental dimensions, indicating that managers assume that social dimensions are more important to a decision-making process than environmental ones. Indeed, the social dimensions have a huge impact on operators and their relationship with the process, resources (equipment), and human needs. Thus, the decision-making model from this research clearly shows that human factors have direct impact on organization sustainability, specifically in operational management.

Furthermore, owing to the inseparable TBL dimensions in sustainability analysis, this trade-off analysis shows that social and environmental factors are not as important as economic factors. For sustainability evaluation, this trade-off result is a critical problem for a competitive organization. It implies that this indicator is useful for analyzing further decisions to improve organizational performance.

Task 2 reflects the same results as task 1. Economic factors are more critical than social and environmental factors. Specifically, in this case, social factors are less impacted than environmental factors. This implies that managers should consider that task 2 has higher sensitivity to environment than task 1. The economic analysis is higher. However, human needs are not equally considered in the decision-making process, which identifies possible critical decisions about sustainability.

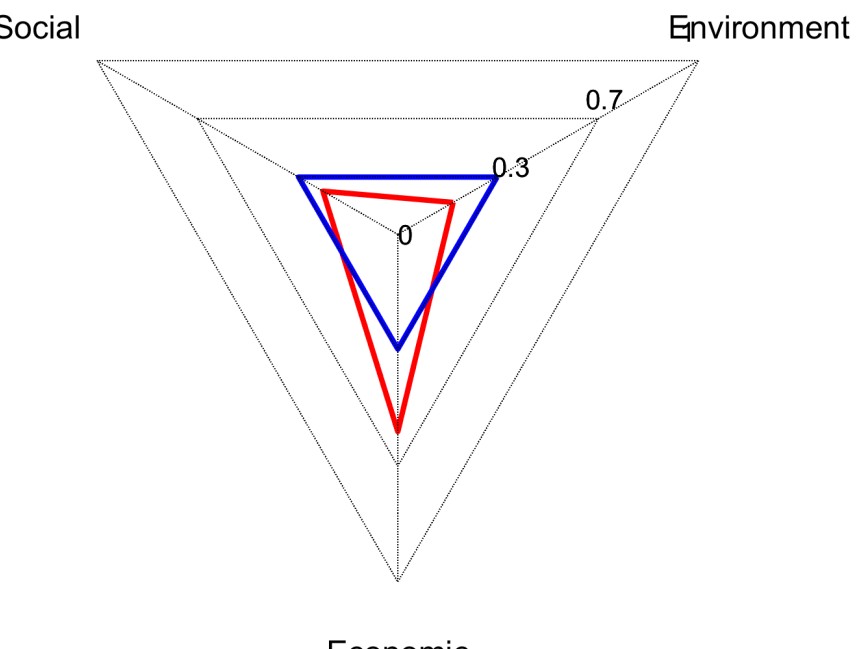

**Figure 10.** Trade-off Analysis of sustainability performance in task 1 (*tv2*).

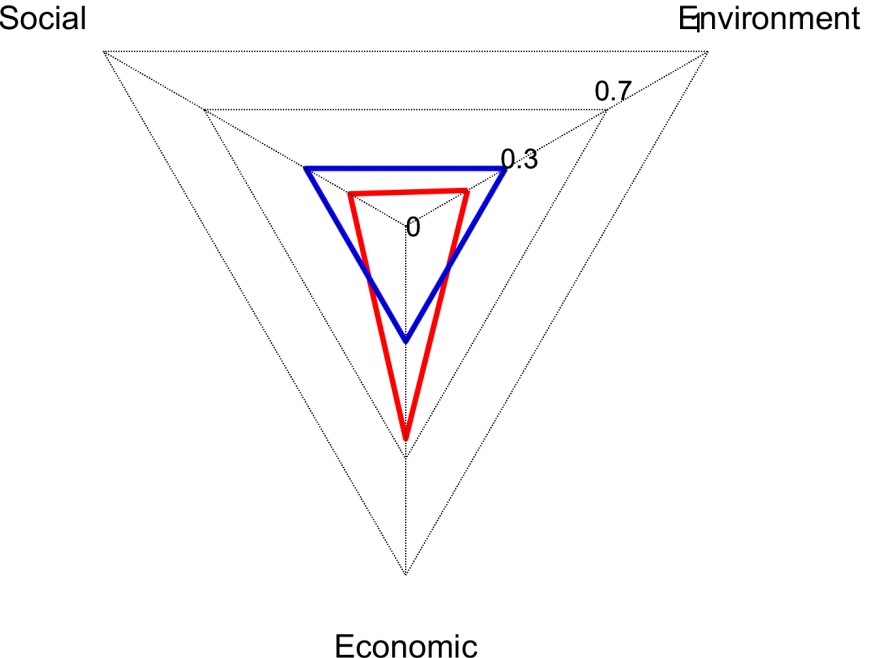

**Figure 11.** Trade-off Analysis of sustainability performance in task 2 (*tv3*).

Table 7 shows the TVM scores of a trade-off analysis based on the decision-support system proposed by the framework integrating TVM and sustainability dimension impacts.

**Table 7.** Trade-off analysis parameters.

| Task | Economic | Social | Environment |
|------|----------|--------|-------------|
| Task 1 | 0.58 | 0.24 | 0.18 |
| Task 2 | 0.60 | 0.18 | 0.22 |

The proposed framework is a powerful tool to guide managers in making decisions related to operational activities and a sustainability approach. The framework can increase the operational decision efficiency of an organization through a holistic view. The results show that environmental and social dimensions have a lower impact over processes. The trade-off analysis shows that social and environmental dimensions require extra attention in pharmaceutical distribution centers and should be constantly evaluated by managers using the proposed framework.

## 4. Conclusions

The sustainability of organizations is necessary to ensure competitiveness in their field and must be evaluated using a holistic view. Specifically, in the operations management level, the TBL should be evaluated to help managers make better decisions. The results of the trade-off analysis of process time variability and TBL sustainability dimensions show that operational managers define their decision-support model with lower impact over social and environmental dimensions in CODSS. The process was validated through a control-chart analysis, identifying the controlled processes for a trade-off analysis. The TVM analysis is more effective if the process is controlled, but it may be used in unbalanced systems as well.

Based on a consolidated framework analysis, this research demonstrates that social and economic dimensions require more attention to improve decision-making and to achieve a more effective and efficient process. Furthermore, the trade-off analysis shows that each task has different impacts on TBL dimensions. Task 1 has more impact in social dimensions, which indicates that human needs should be considered in decisions. Task 2 has more environmental impact in the process; hence, it is evident that sustainability in operations management is essential for organization competitiveness.

As for managerial implications, this research demonstrates that sustainability is essential to improving day-to-day decisions. The trade-off analysis shows how economic dimension drives managers to take operational decisions. First, the control chart shows if the process is controlled. Furthermore, social and environmental impacts are relatively close to economic decisions that show a clear profitability relationship. This study shows that social and environmental dimensions also drive decisions to higher process efficiency and, by consequence, higher profitability. Specifically, in a CODSS, operations managers who do not insert sustainability in their decisions, are not considering dimensions that can cause a direct impact on organization performance.

This study uses the AHP method to define the decision-support model, but the framework allows the use of quantitative and qualitative methods to achieve better results in the trade-off comparison. Thus, future research based on quantitative models and big data analysis should be used to allow managers to make more effective decisions for their organizations.

Furthermore, this framework is validated in a distribution center and is also consistent for discrete production. However, a future research on the continuous production system is hoped to reach a more effective and global framework. The sustainability analysis and costs of organizational operations may be analyzed by this framework, allowing the financial impacts of sustainability in operations management to be defined better.

Another limitation of this study is that it was only applied for time variability, and if it is used with other variability parameters, a mathematical model adjustment must be defined previously in the framework structure to grant a consistent trade-off analysis.

**Author Contributions:** Conceptualization, J.T.d.G.A.A.C., A.M.S.F., F.G.M.F.; methodology, J.T.d.G.A.A.C., R.d.C.V., A.M.S.F. and F.G.M.F.; validation, F.G.M.F., A.M.S.F. and F.G.M.F.; writing—original draft preparation, J.T.d.G.A.A.C.; writing—review and editing, J.T.d.G.A.A.C. and F.G.M.F. All authors have read and agreed to the published version of the manuscript.

**Funding:** This research received no external funding.

**Conflicts of Interest:** The authors declare no conflict of interest. The funders had no role in the design of the study; in the collection, analyses, or interpretation of data; in the writing of the manuscript, or in the decision to publish the results.

## Abbreviations

The following abbreviations are used in this manuscript:

| | |
|---|---|
| TBL | Triple-bottom-line |
| DSS | Decision-support system |
| AHP | Analytical hierarchy process |
| TVM | Time variability management |
| SSCM | sustainable supply chain management |
| UCL | Upper control limit |
| LCL | Lower control limit |
| TVMF | Time variability management factor |
| CR | Consistency Ratio |
| CODSS | Customer orders drug separation |

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
