# Peer review of "Operational Decisions and Sustainability: A Brazilian Case of a Drugs Distribution Center"

_sustainability, doi:10.3390/su12218916_

Round 1
Reviewer 1 Report
Comments:
- The information presented in this article is new and suitable for the academic journal’s target.
- However, the paper is not well structured, and the authors need to clarify and point out relevant studies in the literature review.
- Although the data supports the research findings and conclusions, but this manuscript must be compared with other studies.
- The discussion part and conclusions must be improved to clarify the contributions of those results to sustainability literature
- Finally, the language technical and communication must be improved; this was quite a difficult article to assess because of English errors.
Reviewer 2 Report
This paper aims to evaluate the sustainability of the operations of a drugs distribution center using analytical methods to evaluate operational activities.
General comments:
- A more in-depth review of the existing literature would be appropriate. Since the analysis aims to use the TBL model, it is necessary to expand the literature on it and, more generally, on the concept of sustainability. In addition, in line 66 the authors state that the use of sustainability performance measurement systems in operations management is still unexplored. However, there are some studies in the literature that the authors could cite as reference, for example Johansson, A., Gustavsson, L., & Pejryd, L. (2020). Sustainable operations management through development of unit cost performance measurement. Procedia Manufacturing, 43, 344-351.
- In line 42 the authors argue that "strategic levels and sustainability studies are widely explored". Some of these studies should be mentioned.
- At the end of the introductory paragraph, the authors should go deeper into how the paper is articulated, explaining what each section deals with in order to guide the reader in better interpreting the paper.
- Since such a research surely has practical implications, it would be appropriate to implement the conclusions by adding a section on managerial implications.
- The bibliography must be modified: the first citation that appears must be the number [1], the second the number [2] and so on.
Edits:
line 27 – suggest rewording to “[…] to customers is highly […]”
line 96 – substitute base with based
Reviewer 3 Report
1.Does the 1 in eq. (2) confirm its definition?
2.Table 1 needs to explain in more detail the impact of the rise and fall of its Idle Time and Activity Time.
3.Figure 2 needs more literature support.
4.Page 6, line 181, "@@@Furthermore", needs to confirm its content?
5.Lack of narrative about future research directions.
Reviewer 4 Report
The authors propose a framework oriented to the tradeoff analysis of process time variability and triple-bottom-line (TBL) sustainability dimensions for the supply chain management in the pharmaceutical sector. The framework is validated using data from a Brazilian drugs distribution centre.
Although the proposed model needs to be validated more comprehensively and exhaustively, the work presented in the manuscript is interesting. No significant methodological errors are perceived in the paper. There are some typographical errors in the manuscript (which should be amended in the paper layout process).
Author Response
Thank you for the comment. The paper was revised to avoid typographical errors.
Round 2
Reviewer 1 Report
1) The manuscript's organization is appropriate, and the overall highlight of the results has clearly been presented.
2) Last but not least, the language technical and communication is good, and the adequacy of writing is up to the mark but can be improved
I hope that these comments can help improve the manuscript. Good luck!